# DAPK interacts with Patronin and the microtubule cytoskeleton in epidermal development and wound repair

**Marian Chuang, Tiffany I Hsiao, Amy Tong, Suhong Xu†, Andrew D Chisholm***

Section of Cell and Developmental Biology, Division of Biological Sciences, University of California, San Diego, San Diego, United States

**Abstract** Epidermal barrier epithelia form a first line of defense against the environment, protecting animals against infection and repairing physical damage. In *C. elegans,* death-associated protein kinase (DAPK-1) regulates epidermal morphogenesis, innate immunity and wound repair. Combining genetic suppressor screens and pharmacological tests, we find that DAPK-1 maintains epidermal tissue integrity through regulation of the microtubule (MT) cytoskeleton. *dapk-1* epidermal phenotypes are suppressed by treatment with microtubule-destabilizing drugs and mimicked or enhanced by microtubule-stabilizing drugs. Loss of function in *ptrn-1*, the *C. elegans* member of the Patronin/Nezha/CAMSAP family of MT minus-end binding proteins, suppresses *dapk-1* epidermal and innate immunity phenotypes. Over-expression of the MT-binding CKK domain of PTRN-1 triggers epidermal and immunity defects resembling those of *dapk-1* mutants, and PTRN-1 localization is regulated by DAPK-1. DAPK-1 and PTRN-1 physically interact in co-immunoprecipitation experiments, and DAPK-1 itself undergoes MT-dependent transport. Our results uncover an unexpected interdependence of DAPK-1 and the microtubule cytoskeleton in maintenance of epidermal integrity.

**\*For correspondence:** chisholm@ucsd.edu

**Present address:** †School of Medicine, Zhejiang University, Hangzhou, China

**Competing interests:** The authors declare that no competing interests exist.

## Introduction

Death-associated protein kinase 1 (DAPK1) and its related calcium-regulated serine/threonine kinases play a wide variety of roles in cell death and tumor suppression (*Bialik and Kimchi, 2006*). Mammalian DAPK1 has been implicated in stress responses (*Tu et al., 2010*), antiviral immunity (Zhang et al.), and in IL-1ß-associated inflammatory diseases (*Chakilam et al., 2013*; *Chuang et al., 2011*). In addition, DAPK1 can act as a checkpoint in the macrophage inflammation program (*Mukhopadhyay et al., 2008*) and as a negative regulator of T-cell receptor-mediated activation of NFκB (*Chuang et al., 2008*).

In the nematode *C. elegans* the sole DAPK family member, DAPK-1, plays roles in autophagy and in excitotoxic neuronal death (*Del Rosario et al., 2015*; *Kang and Avery, 2010*). DAPK-1 also regulates epidermal development and wound repair, independently of known cell death programs. The nematode epidermis is a barrier epithelium that forms the first line of defense against environmental stresses, such as pathogens and physical damage (*Engelmann and Pujol, 2010*). Mutations in *dapk-1* result in progressive degeneration of the epidermis, cuticle hypertrophy, and constitutive activation of epidermal innate immune responses via a p38 MAPK cascade (*Tong et al., 2009*). Such *dapk-1* mutants behave as if they are constitutively wounded even without injury, and when wounded exhibit faster wound repair (*Xu and Chisholm, 2011*). How DAPK-1 regulates these multiple aspects of epidermal maintenance and wound repair is not yet understood.

Here we took a genetic approach to understanding DAPK-1's functions in the epidermis. We identified genetic suppressors and enhancers of *dapk-1* morphological defects, revealing novel roles

for microtubule (MT) regulators in the mature epidermis. Genetic and pharmacological manipulations suggest aberrant *dapk-1* function causes excessive MT stabilization, resulting in morphological defects. *dapk-1* epidermal defects can be suppressed by loss of function in the MT minus end binding protein PTRN-1 and by pharmacological destabilization of MTs. Moreover, overexpression of the MT-binding domain of PTRN-1 is sufficient to induce *dapk-1*-like epidermal defects. Our data suggest DAPK-1 destabilizes epidermal MTs by inhibiting the function of PTRN-1. We further show for the first time that DAPK-1 itself undergoes MT-dependent transport. Our findings reveal an unexpected interplay between DAPK-1, the epithelial MT cytoskeleton, and epidermal morphology and wound repair.

## Results

### *dapk-1* epidermal morphological defects can be suppressed or enhanced by loss of function in microtubule regulators

To identify new *dapk-1* interactors, we screened for genetic suppressors of *dapk-1* epidermal phenotypes. All *dapk-1* mutants display epidermal morphological (Mor) defects, with penetrance varying depending on the allele (*Figure 1A*, *Figure 1—figure supplement 1A*) (*Tong et al., 2009*). The Mor phenotype reflects a progressive accumulation of the cuticle and degeneration of the underlying epidermis at the extreme anterior and posterior, as well as the dorsal midline. *dapk-1(ju4)*, which causes a missense alteration S179L in the DAPK-1 kinase domain, causes 100% of animals to display this aberrant morphology. We mutagenized *dapk-1(ju4)* animals and screened for suppression of the Mor phenotype (see Methods). We identified multiple extragenic suppressors of *dapk-1(ju4)*, two of which are described here. One suppressor, *ju698*, causes a nonsense mutation in *ptrn-1*, which encodes the *C. elegans* member of the Patronin/CAMSAP/Nezha family of MT minus end binding proteins. A null allele *ptrn-1(lt1)* suppressed *dapk-1(ju4)* phenotypes to the same extent as *ptrn-1(ju698)* (*Figure 1B*; *Figure 1—figure supplement 1C*). Suppression of *dapk-1(ju4)* by *ptrn-1(0)* was rescued by a single-copy insertion (Mos-SCI) *ptrn-1(+)* transgene and by transgenes expressing PTRN-1 under the control of the *dpy-7* promoter, specific to the larval epidermis, indicating that loss of PTRN-1 function in the larval epidermis is required for suppression of the *dapk-1(ju4)* phenotype.

Our screen also identified a mutation in the dynein heavy chain, *dhc-1(ju697)*, which results in a missense change G2537S in the fifth P-loop of DHC-1. Complete loss of function in *dhc-1* causes lethality (*Gonczy et al., 1999*); an independent loss-of-function allele *dhc-1(or195ts)* fully suppressed *dapk-1(ju4)* morphological defects at 20°C (*Figure 1B*; *Figure 1—figure supplement 1C*). This suppression was rescued by a DHC-1::GFP single-copy insertion transgene (*Figure 1B*). PTRN-1 and other CAMSAP proteins interact with MT minus ends (*Goodwin and Vale, 2010*; *Jiang et al., 2014*), and dynein is critical for MT minus-end dependent transport and MT organization. We found that *dhc-1(or195)* or *dhc-1(or195); ptrn-1(0)* caused more complete suppression of *dapk-1(ju4)* phenotypes than did *ptrn-1(0)*, suggesting DHC-1 might affect additional pathways required for *dapk-1* function.

We also isolated two intragenic suppressors, *ju1143* and *ju1145*, which result in nonsense mutations (Q608stop and R48stop) in *dapk-1(ju4)* (*Figure 1—figure supplement 1B*). These mutations reduce the penetrance of *dapk-1(ju4)* Mor phenotypes from 100% to 40% and 30% respectively, and resemble the *dapk-1(gk219)* deletion allele and other newly generated deletions of the *dapk-1* locus (*Figure 1—figure supplement 1B,D*). Although *dapk-1(ju4)* phenotypes are recessive, the identification of such intragenic suppressors suggests *ju4* causes a gain of function (see Discussion). Both *ptrn-1(lt1)* and *dhc-1(or195)* completely suppressed *dapk-1(gk219)* morphological phenotypes (*Figure 1B*), suggesting loss of function in PTRN-1 or DHC-1 bypasses the requirement for DAPK-1.

Based on our identification of two MT-interacting proteins in our suppressor screen we tested additional MT-associated factors, as well as orthologs of genes known to interact with DAPK1 or CAMSAPs (*Table 1*). Partial loss of function in *unc-116*, which encodes a kinesin-1 plus-end directed motor, suppressed *dapk-1(ju4)* morphological defects. DAPK family members have previously been implicated in MT stability, promoting function of the MT-associated protein tau via the Pin1 prolyl isomerase or the MARK kinase (*Kim et al., 2014*; *Wu et al., 2011*). However, loss of function in orthologs of these genes (*ptl-1*, *pinn-1*, *par-1*) did not modify *dapk-1(ju4)* phenotypes (*Table 1*), suggesting DAPK-1 regulates epidermal MTs via a novel mechanism.

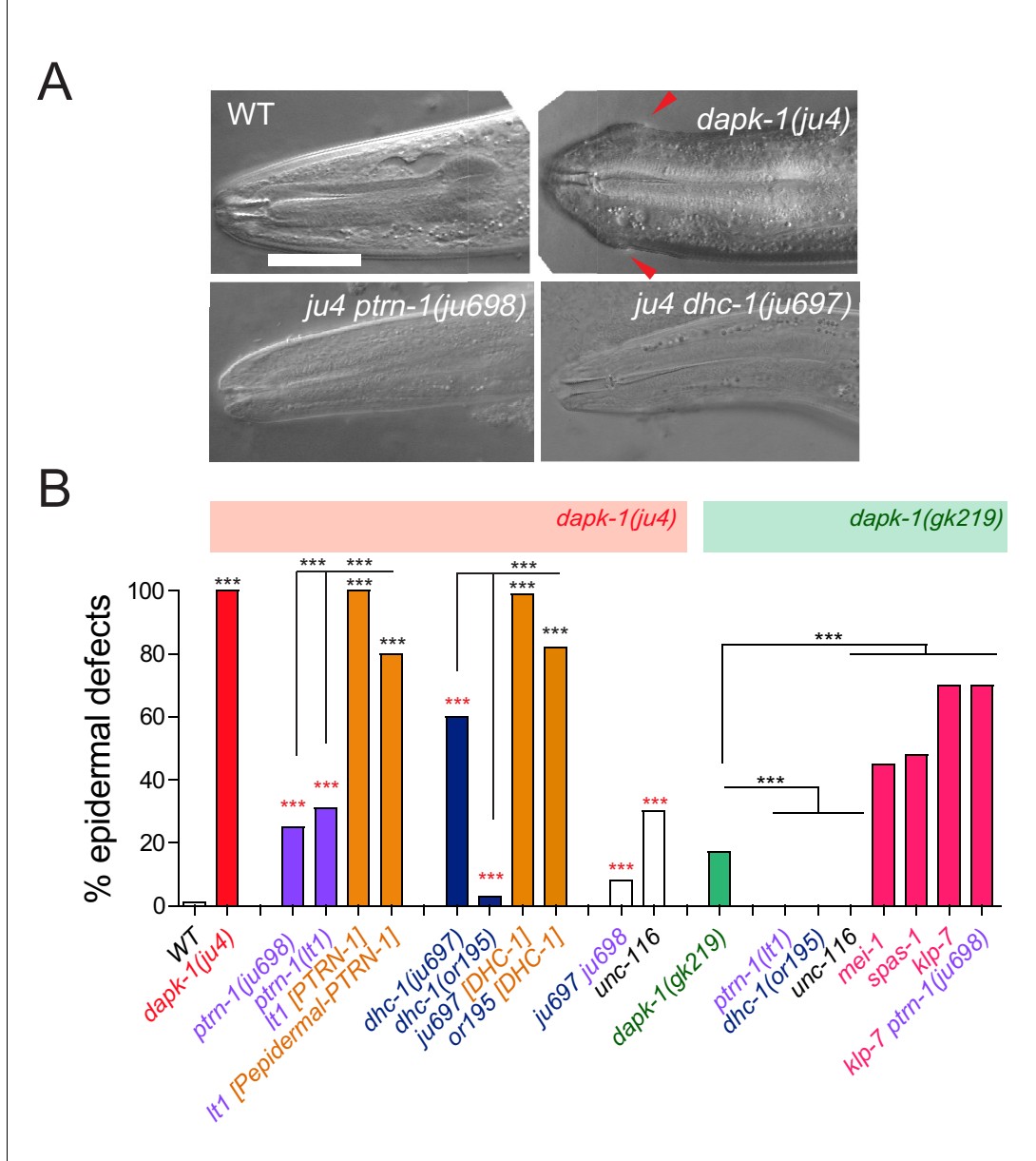

Figure 1. *dapk-1* epidermal phenotypes are modified by mutations in cytoskeletal regulating genes. (A) Anterior epidermal morphology of wild type adult and aberrant morphology (Mor phenotype, red arrowhead) of *dapk-1(ju4)* animals; suppression by *ptrn-1(ju698)* and *dhc-1(ju697)*. DIC images; scale bar, 10 μm. Anterior is to the left and dorsal up. (B) Loss of function in *ptrn-1* or *dhc-1* suppresses the anterior epidermal morphology defects of *dapk-1(ju4)* and *dapk-1(gk219)*. Mor penetrance at 20°C; N>100 per genotype. Fisher's exact test; ***p<0.001. Black stars in B are comparisons to WT; red stars are comparisons to *dapk-1(ju4)*

The following figure supplement is available for figure 1:

Figure supplement 1. Partial suppression of *dapk-1* phenotypes by loss of function in cytoskeletal regulators

CAMSAPs are thought to promote MT stability in part by protecting minus ends from the MT depolymerizing enzyme kinesin-13 (*Goodwin and Vale, 2010*). Consistent with this model, loss of function in *klp-7*/kinesin-13 strongly enhanced morphological defects of *dapk-1(gk219)*, both in *ptrn-1(+)* and in *ptrn-1(0)* backgrounds (*Figure 1B*). Partial loss of function of the MT severing enzymes MEI-1/p60 katanin or SPAS-1/Spastin also enhanced *dapk-1(gk219)* Mor phenotypes. None of these mutants conferred morphological defects in a *dapk-1(+)* background (*Figure 1—figure supplement*

**Table 1.** Suppressors and enhancers of *dapk-1* morphological defects.

| Gene | Alleles and sequence change | | Mammalian orthologs |
|---|---|---|---|
| A. Suppressors (forward screen) | | | |
| ptrn-1 | ju698 | | CAMSAP/Patronin |
| | *lt1 | | |
| | tm5597 | | |
| dhc-1 | ju697 | | Dynein heavy chain |
| | *or195ts | | |
| dapk-1 | ju1143 | | (intragenic) |
| | ju1145 | | |
| B. Suppressors (candidates) | | | |
| unc-116 | e2310 | | Kinesin-1 |
| sma-1 | e30 | | Beta-heavy spectrin |
| C. Enhancers | | | |
| klp-7 | tm2143 | | Kinesin-13 |
| mei-1 | or642ts | | p60 katanin |
| spas-1 | tm683 | | Spastin |
| cat-4 | tm773 | | GTP cyclohydrolase I |
| F47G4.5 | ok2667 | | p80 katanin |
| D. No interaction | | | |
| ebp-1 | tm1357 | | Plus-end binding protein |
| ebp-2 | gk756 | | Plus-end binding protein |
| ccpp-1 | ok1821 | | Cytosolic Carboxypeptidase |
| ccpp-6 | ok382 | | Cytosolic Carboxypeptidase |
| mcrs-1 | tm3681 | | Microspherule Protein 1 |
| efa-6 | tm3124 | | EFA6 |
| ttll-5 | tm3360 | | Tubulin tyrosine ligase-like |
| ttll-11 | tm4059 | | Tubulin tyrosine ligase-like |
| ttll-12 | tm4957 | | Tubulin tyrosine ligase-like |
| unc-70 | e524 | | β-Spectrin |
| dylt-2 | gk762 | | Dynein light chain |
| dnc-1 | or404ts | | p150 dynactin |
| nud-1 | ok552 | | NDE1/NDEL1 |
| nud-2 | ok949 | | NDE1/NDEL1 |
| unc-14 | e57 | | kinesin-1 adaptor |
| tbg-1 | t1465 | | γ-tubulin |
| ptl-1 | ok621 | | tau |
| pinn-1 | tm2235 | | Pin1 |
| par-1 | zu310ts | | MARK |

Suppressors indicated * were tested for suppression of *dapk-1(ju4)* and *dapk-1(gk219)*. Enhancers were tested with *dapk-1(gk219)*. Genes in section D were mostly tested for interaction with *dapk-1(ju4)*. *dapk-1 tbg-1* double mutants were extremely sick, and a stable strain could not be obtained; n > 100 animals scored per genotype.

*1D*). These analyses suggest that aberrant *dapk-1* function causes epidermal integrity to be sensitized to MT stability. *dapk-1(ju4)* defects were not suppressed by loss of function in several other genes implicated in MT dynamics (*Table 1*), suggesting a specific subset of MT regulators can affect epidermal morphogenesis.

## PTRN-1/Patronin is required for the upregulated epidermal innate immune responses and accelerated wound repair in *dapk-1(ju4)* mutants

In *C. elegans* sterile wounding or fungal infection induces expression of antimicrobial peptides (AMPs) in the epidermis (*Couillault et al., 2004*; *Pujol et al., 2008*). *dapk-1(ju4)* animals constitutively express high levels of AMPs such as *nlp-29* and *nlp-30*, and this hyperactive epidermal immune response is genetically separable from *dapk-1(ju4)* epidermal morphological defects (*Tong et al., 2009*). To address whether the extragenic suppressors or enhancers of *dapk-1* morphological defects also interacted with the innate immune response, we examined *nlp-29* expression in *dhc-1 (or195)* and *unc-116(e2310)*. Neither mutant suppressed the elevated innate immune response in *dapk-1(ju4)* (*Figure 2A*; *Figure 2—figure supplement 1B*), as measured using a P*nlp-29*-GFP transcriptional reporter. In contrast, *ptrn-1(0)* fully suppressed the elevated immune response of *dapk-1 (ju4)* (*Figure 2A,B*). Among all suppressors tested, only PTRN-1 interacted with DAPK-1 in epidermal morphology and in innate immunity. Given the unique and specific genetic interactions of *ptrn-1* and *dapk-1*, we focused our analysis on PTRN-1.

DAPK-1 acts in the epidermis to maintain epidermal morphology (*Tong et al., 2009*). Although epidermal development is overtly normal in *ptrn-1(0)* mutants, PTRN-1 acts redundantly with γ-tubulin and the ninein-like protein NOCA-1 to assemble noncentrosomal MT arrays essential for epidermal development (*Wang et al., 2015*). However, loss of function in γ-tubulin/TBG-1 did not suppress *dapk-1(ju4)* epidermal defects (*Table 1*), suggesting DAPK-1 specifically interacts with the PTRN-1 pathway.

*dapk-1* mutants also display accelerated epidermal wound closure, manifested by the rate of closure of actin rings that form around puncture wounds (*Xu and Chisholm, 2011*). We found that *ptrn-1* mutations suppressed this accelerated wound closure to wild type rates (*Figure 2C,D*, *Figure 2—figure supplement 1C,D*). Moreover, *ptrn-1(0)* single mutants displayed significantly delayed wound closure compared to the wild type. Thus, DAPK-1 and PTRN-1 play antagonistic roles in epidermal development and in wound repair.

## Pharmacological modulation of MT stability can suppress or enhance *dapk-1* morphological defects

Because the *dapk-1* epidermal phenotypes are suppressed by loss of function in a MT stabilizing factor (Patronin/*ptrn-1*) and enhanced by loss of function in MT destabilizing factors (Kinesin-13/*klp-7*, Katanin/*mei-1*, Spastin/*spas-1*) we hypothesized that epidermal defects in *dapk-1* mutants might result from excessive stabilization of epidermal MTs. We therefore tested whether drugs that depolymerize MTs (colchicine, nocodazole) or stabilize MTs (paclitaxel) could modify epidermal Mor defects. Some experiments used *cat-4* mutants, which are defective in biopterin synthesis and display leaky cuticles and hypersensitivity to drugs (*Loer et al., 2015*).

Colchicine treatment significantly suppressed the epidermal morphology defects of *dapk-1 (gk219)* animals (*Figure 3A*). *cat-4* itself strongly enhanced *dapk-1(gk219)*, suggesting that *dapk-1* is also sensitized to defects in cuticle integrity (*Figure 3—figure supplement 1A*). The morphological defects of *dapk-1(gk219) cat-4* animals were suppressed by colchicine in a dose-dependent manner, and high concentrations of colchicine (5 mM) significantly suppressed *dapk-1(ju4)* phenotypes (*Figure 3—figure supplement 1A*). Conversely, treatment with the MT stabilizing drug paclitaxel enhanced *dapk-1(gk219)* morphological defects, consistent with Mor phenotypes in *dapk-1(ju4)* mutants being caused by hyper-stabilized MTs (*Figure 3A*). Indeed, paclitaxel treatment of wild type animals could induce *dapk-1*-like epidermal defects, albeit at very low penetrance (*Figure 3A,B*).

*ptrn-1(0)* mutants display overtly normal epidermal morphology, but were six times more sensitive to the effects of paclitaxel compared to WT animals (*Figure 3A,B*). Paclitaxel also induced morphological defects in *dapk-1(gk219) ptrn-1(0)* double mutants (*Figure 3—figure supplement 1A*), although not to the same extent as in *dapk-1(gk219)* single mutants (*Figure 3A*). *dhc-1(or195)* single mutants were also hypersensitive to paclitaxel, compared to wild type (*Figure 3—figure*

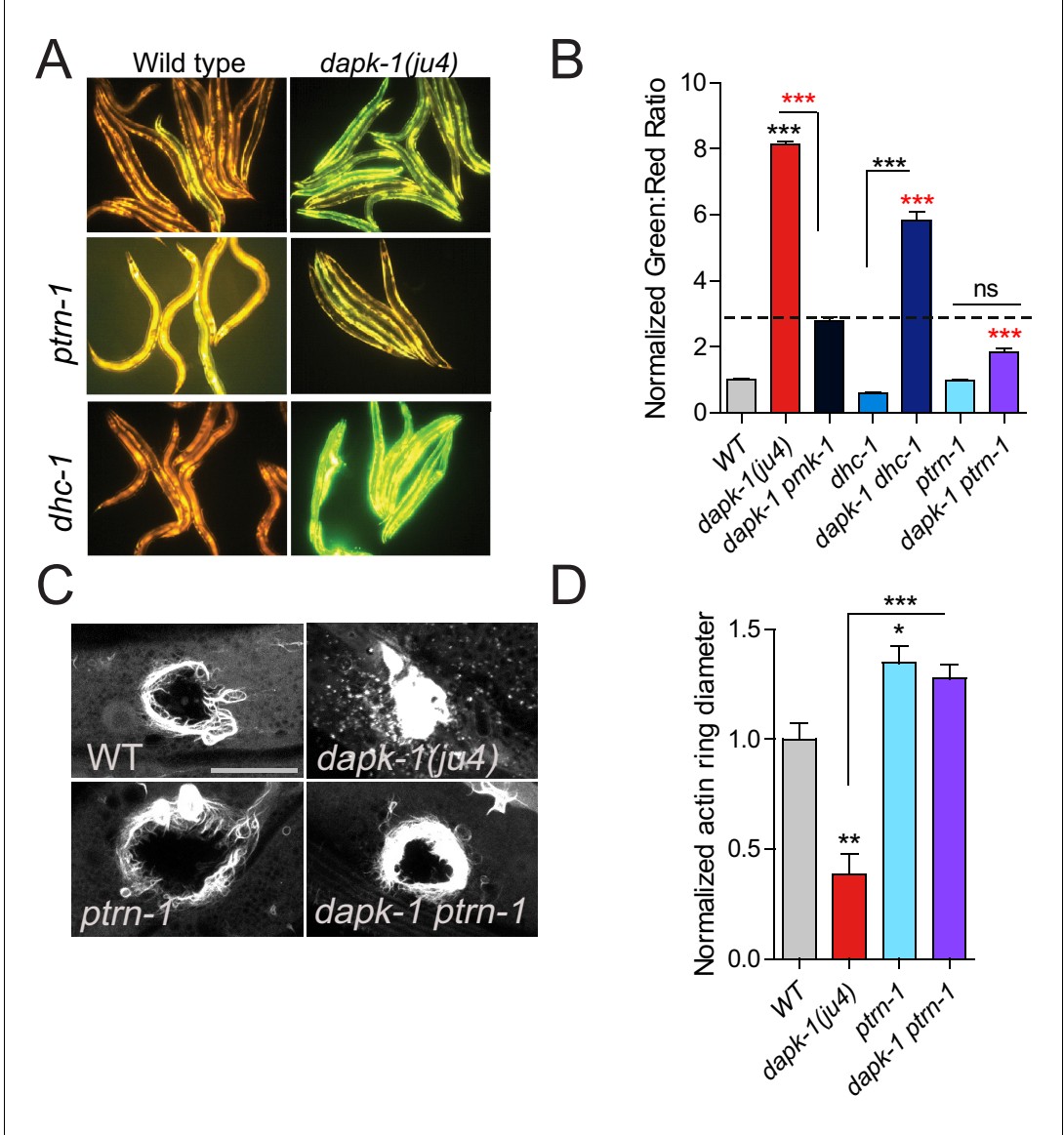

**Figure 2.** *ptrn-1* suppresses *dapk-1(ju4)* innate immune and wound repair phenotypes. (A) *dapk-1(ju4)* animals display an elevated expression of the antimicrobial peptide *nlp-29* (P*nlp-29*-GFP, *frIs7*); this is suppressed by *ptrn-1(0)*, whereas *dhc-1(or195)* and other mutations do not suppress (see also *Figure 2—figure supplement 1A*). Day 1 adults. (B) Quantitation of fluorescence intensity ratio (Green:Red, normalized to WT = 1) of animals in A, using the COPAS Biosort. N>100. Fisher's exact test; ***p<0.001. *pmk-1* is a MAPK required for activation of *nlp-29* transcription; *pmk-1* mutants serve as a negative control. (C) *ptrn-1(0)* suppresses the accelerated actin ring closure in *dapk-1(ju4)* mutants after needle wounding. The wound-triggered actin ring is visualized using P*col-19*-GFP::moesin (*juIs352*); scale, 10 μm. (D) Quantitation of actin ring diameter, mean ± SEM. One-way ANOVA and Dunn's post-test; *p<0.05; **p<0.01; ***p<0.001.

The following figure supplement is available for figure 2:

**Figure supplement 1.** Most *dapk-1* modifiers do not affect the *dapk-1(ju4)* constitutively active innate immune response.

supplement 1A), possibly reflecting increased free tubulin concentration in these mutants. Taken together, these data are consistent with the model that *dapk-1* mutants display excessively stabilized MTs that cause aberrant epidermal morphology.

We also asked whether pharmacological manipulation of MTs affected the epidermal innate immune response. At high concentrations of colchicine we noted significant suppression of *dapk-1 (ju4)* innate immune responses. Conversely, 5 μM paclitaxel induced P*nlp-29*-GFP expression in 40%

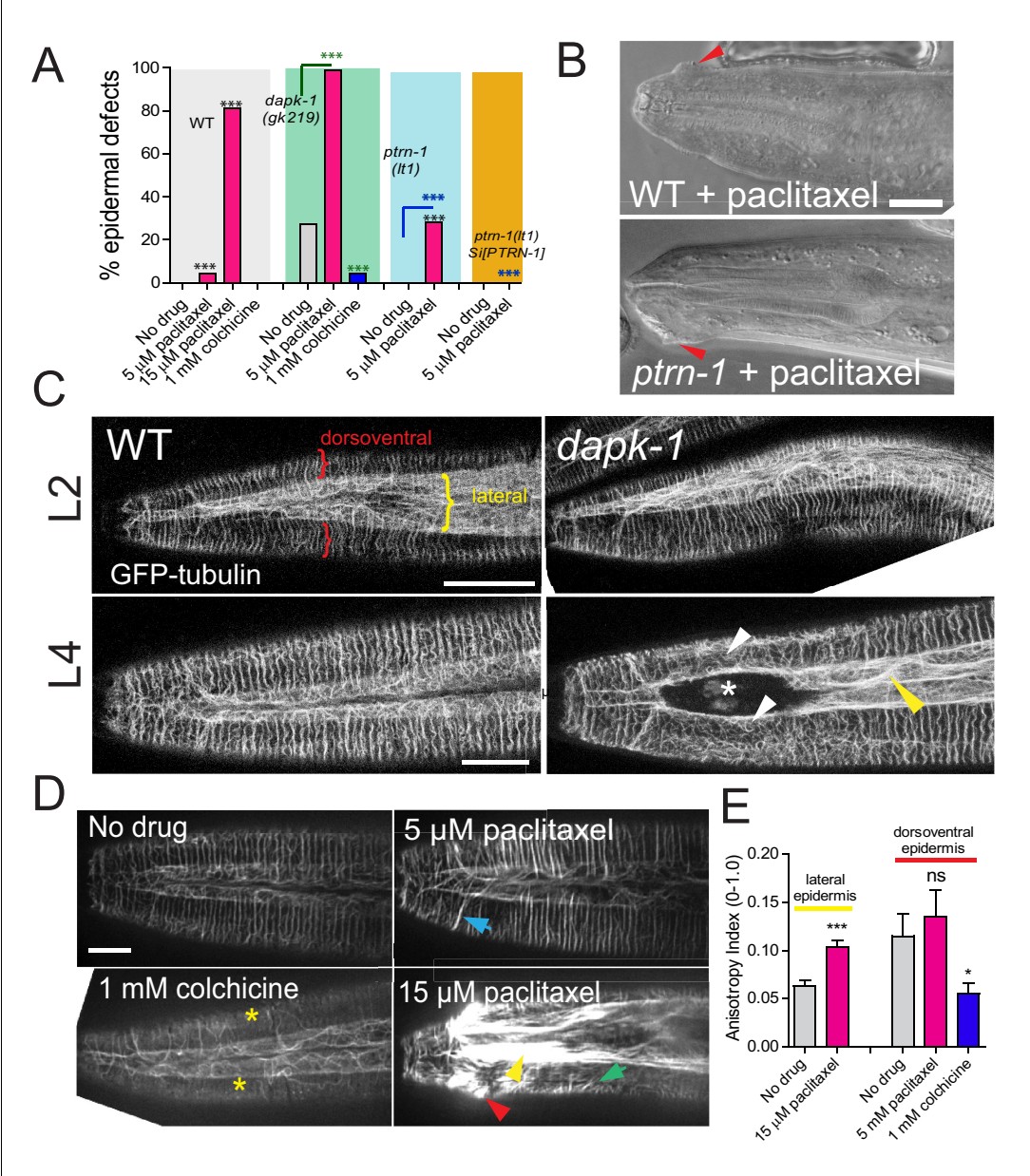

**Figure 3.** *dapk-1* epidermal morphology defects are mimicked or enhanced by MT stabilization and suppressed by MT depolymerization. (**A**) MT stabilization by paclitaxel treatment induces *dapk-1*-like morphological defects in the wild type and in *ptrn-1(0)* mutants, and enhances the morphological defects of *dapk-1(gk219)*. Colchicine treatment suppresses *dapk-1(gk219)* morphological defects. Statistics: Fisher's exact test; ***p<0.001; n > 100 animals per condition. (**B**) DIC images of epidermal defects in representative wild type and *ptrn-1(0)* animals after paclitaxel treatment. (**C**) Confocal images of epidermal MT architecture in the heads of WT and *dapk-1(ju4)* L2 and L4 larvae (P*dpy-7*-GFP::TBB-2, *ltSi570*). Yellow bracket indicates lateral epidermal ridge, red brackets mark dorsoventral epidermis overlying muscles. *dapk-1(ju4)* L4 animals display bundling of MTs in the lateral epidermis (yellow arrowhead) and disorganized circumferential MTs (white arrowhead). (**D**) Effects of MT drugs on epidermal MT organization (*ltSi570*) in heads of L4 animals. Treatment with colchicine causes loss of circumferential MT bundles (yellow asterisks). Treatment with low concentrations of paclitaxel (5 μM) causes circumferential bundles to be brighter and straighter than in wild type (blue arrow); higher concentrations of paclitaxel (15 μM) induce overt MT bundling (yellow arrow), crossing of circumferential MTs (green arrow), and *dapk-1*-like morphological defects (red arrowhead). All scale bars, 10 μm. (**E**) Treatment with 15 μM paclitaxel significantly increases the anisotropy of MT bundles in the lateral epidermis. Statistics, Kruskal-Wallis and Dunn's post test; *p<0.05, ***p<0.001; N>8 animals.

The following figure supplement is available for figure 3:

**Figure supplement 1.** Effects of MT-altering drugs on epidermal morphology and innate immune responses in mutant backgrounds.

of *ptrn-1(0)* mutants, including in animals not displaying strong Mor phenotype (*Figure 3—figure supplement 1B*), while higher concentrations of paclitaxel (15 μM) could induce P*nlp-29*-GFP in a subset of WT animals. Thus, destabilization of epidermal MTs can reverse the upregulation of epidermal innate immune responses in *dapk-1* mutants, while hyper-stabilization of MTs is sufficient to induce the innate immune response, possibly by compromising epidermal integrity.

## *dapk-1* mutants display aberrant MTs resembling those of paclitaxel-stabilized animals

The above analyses suggest *dapk-1* mutants might display excessively stabilized MTs. We therefore analyzed epidermal MT architecture using the tubulin marker TBB-2::GFP (see Materials and methods; *Figure 4—figure supplement 1A*). In the lateral hyp7 epidermal syncytium MTs form a dense meshwork mostly oriented along the anteroposterior axis. In contrast, the dorso-ventral compartment of hyp7, overlying body wall muscles, contains parallel circumferential MT bundles spaced 1–1.5 μm apart (*Figure 3C*; *Figure 4—figure supplement 1B,C*; *Figure 4E*). These arrays extend from the lateral epidermis to the dorsal or ventral midlines.

Chronic treatment with colchicine caused breakage and loss of circumferential MT bundles in the dorsoventral epidermis (*Figure 3D*); in the lateral compartment, MTs became sparse and less bundled than in the WT. Conversely, paclitaxel treatment caused circumferential MT bundles to be straighter and thicker (*Figure 3D,E*). At high paclitaxel concentrations (15 μM), circumferential MT bundles became disorganized and lateral MT bundles were thicker (*Figure 3D,E*). Thus, MT stabilization by paclitaxel results in MTs becoming straighter and more bundled.

We then examined MTs in *dapk-1(ju4)* mutants, focusing on the anterior epidermis, where *dapk-1* morphological defects begin. In early (L2) larvae the MT architecture of *dapk-1(ju4)* animals appeared normal (*Figure 3C*; *Figure 4—figure supplement 1B,C*). By the late larval (L4) stage a region devoid of MTs appeared in the anterior lateral epidermis of *dapk-1(ju4)* animals (*Figure 3C*, asterisk), apparently due to local degeneration of the epidermis and cuticle overgrowth (*Tong et al., 2009*). Nearby circumferential MT bundles were disorganized (*Figure 3C*, white arrowheads), however most circumferential MT bundles appeared normal (*Figure 4—figure supplement 1B,C*), suggesting the local MT defects might be a secondary consequence of the epidermal degeneration. In *dapk-1(ju4)* adults MT bundles were spaced normally (*Figure 4E,F*) but were thinner (*Figure 4—figure supplement 1C*). In the lateral epidermis of *dapk-1(ju4)* animals MT bundles were straighter and more bundled than in wild type (*Figure 4A*, yellow arrowheads), resembling those of paclitaxel-treated animals. We used FibrilTool to measure the anisotropy (directional dependence) of lateral MT bundles (see Methods) and found that *dapk-1(ju4)* mutants displayed increased MT bundle anisotropy compared to wild type (*Figure 4C*). The MT defects of *dapk-1(gk219)* mutants were very similar to those of *dapk-1(ju4)* (*Figure 4—figure supplement 1D–H*). The similarities between the *dapk-1* mutant phenotypes and the effects of paclitaxel treatment suggest DAPK-1 normally destabilizes epidermal MTs.

We next asked whether loss of function in PTRN-1 affected the disorganization of MT architecture in *dapk-1(ju4)*. *ptrn-1(0)* mutant larvae have slightly fewer circumferential MT bundles (*Wang et al., 2015*). We found that *ptrn-1(0)* adults displayed a significant loss of circumferential MT bundles (~50% of WT; *Figure 4A,E,F*); remaining bundles were shorter than in the wild type (*Figure 4D*). Lateral MT bundles in *ptrn-1(0)* adults displayed normal anisotropy (*Figure 4C*). In *dapk-1(ju4) ptrn-1(0)* double mutants the number and length of circumferential MT bundles resembled that of *ptrn-1(0)* mutants (*Figure 4D–F*). However *ptrn-1(0)* suppressed the increased anisotropy of lateral MTs in *dapk-1(ju4)* to wild type levels (*Figure 4C*). These results indicate that antagonistic interactions between PTRN-1 and DAPK-1 balance MT architecture in the epidermis.

We next assessed the effects of *dapk-1* mutants on epidermal MT dynamics using the MT plus end marker EBP-2::GFP (EBP-GFP for brevity). EBP-GFP binds to growing MT plus ends and in vivo forms moving comets whose movement can be quantified by kymograph analysis (see Methods). We found that MT plus ends in the wild type adult epidermis were highly dynamic, with $0.09 \pm 0.015$ comets/$\mu m^2$ in the lateral epidermis, and slightly more ($0.12 \pm 0.015$ comets/$\mu m^2$) in the dorsoventral epidermis (*Videos 1,2*). The overall density of comets in *ptrn-1(0)* animals was similar to that in wild type (*Figure 4H*), as previously reported for larvae (*Wang et al., 2015*). By contrast, *dapk-1(ju4)* mutants, as well as *dapk-1(ju4) ptrn-1(0)* double mutants, had significantly more comets in the lateral epidermis than in WT or *ptrn-1(0)* (*Videos 3,4*). In wild type adults, EBP-GFP comets grew at $0.35 \pm$

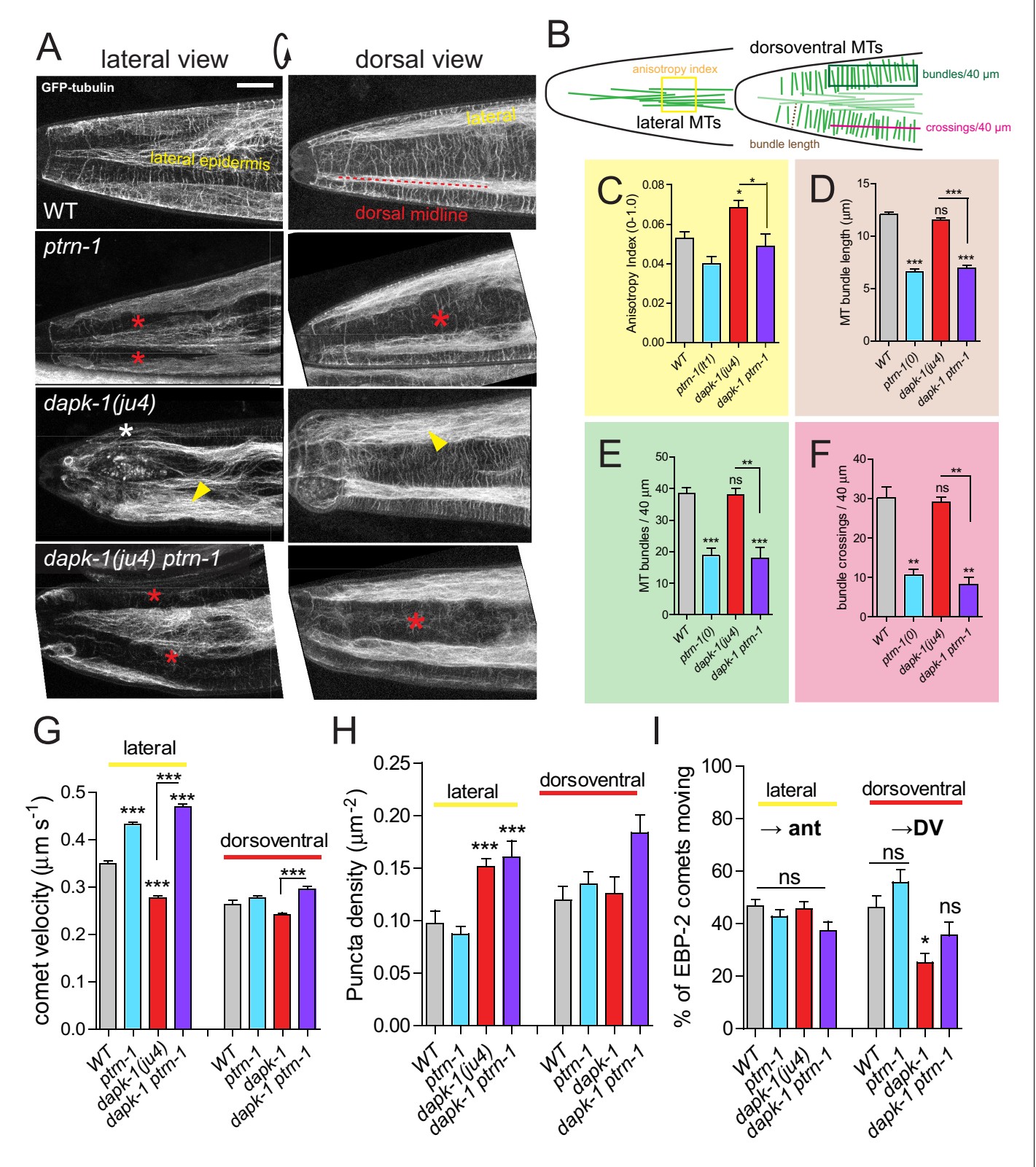

**Figure 4.** *dapk-1* defects in epidermal MT architecture are suppressed by *ptrn-1*. (**A**) *ptrn-1(0)* and *dapk-1(ju4)* display distinct effects on epidermal MT organization; MTs visualized in young adults (P*col-19*-GFP::TBB-2, *juSi239*). Left column, lateral views of the head; right column, dorsal view. *ptrn-1(0)* mutant adults display fewer circumferential MT bundles in the dorsoventral epidermis (red asterisks). *dapk-1(ju4)* mutants display increased bundling in the lateral epidermis (yellow arrowhead), quantified by anisotropy index in panel **C**. *dapk-1(ju4) ptrn-1(0)* double mutants display normal MT bundling in

*Figure 4 continued on next page*

*Figure 4 continued*

the lateral epidermis and reduced dorsoventral MTs (red asterisks). Mor phenotype, white asterisk. Scale, 10 μm. (**B**) Cartoon of MT organization in the *C. elegans* lateral and dorsoventral epidermis; colored boxes indicate ROIs used for quantitation of MT parameters in panels C–F. (**C**) *dapk-1(ju4)* animals display elevated MT bundle anisotropy in the lateral epidermis, which is suppressed by *ptrn-1(0)*. (**D–F**) MT bundle length, density and crossing frequency in the dorsoventral epidermis is reduced in *ptrn-1(0)* and in *dapk-1(ju4) ptrn-1(0)* double mutants. N>8 animals per genotype. Bars show mean ± SEM. Kruskal-Wallis and Dunn's post test; *p<0.05; **p<0.01; ***p<0.001. (**G–I**) Quantitation of epidermal EBP-2::GFP comet dynamics in the lateral and dorsoventral epidermis. N>10 animals per genotype.

The following figure supplement is available for figure 4:

**Figure supplement 1.** Epidermal MTs in wild type and *dapk-1* mutants.

0.006 μm/s in the lateral epidermis, and 0.26 ± 0.01 μm/s in the dorsoventral epidermis, comparable to growth rates in the larval epidermis (*Wang et al., 2015*). MT growth rates in the lateral epidermis were significantly reduced in *dapk-1(ju4)* mutants, and increased in *ptrn-1(0)* single and in *dapk-1 (ju4) ptrn-1(0)* mutants (*Figure 4G*). *dapk-1(ju4)* mutants thus display more slow-growing MT plus ends in the lateral epidermis, consistent with a partial stabilization of MT dynamics.

Finally, we analyzed the directionality of plus-end growth. In the lateral epidermis most EBP-GFP comets grew anteriorly or posteriorly, without a strong bias in directionality. Similarly, in the dorsoventral epidermis equal numbers of EBP-GFP comets grew towards or away from lateral epidermal ridges (*Figure 4I*). EBP-GFP comet directionality was normal in *ptrn-1(0)* mutant and *dapk-1(ju4) ptrn-1(0)* backgrounds. However, *dapk-1(ju4)* mutants displayed a significant bias in comet directionality in the dorsoventral epidermis, such that fewer comets grew away from the lateral epidermis. Thus, in *dapk-1* mutants growing MTs are more confined to the lateral epidermis and less likely to extend into the dorsoventral epidermis.

## Over-expression of the PTRN-1 CKK domain induces *dapk-1*-like morphology defects

To understand how PTRN-1 might regulate epidermal MTs in the *dapk-1* mutant we tested individual PTRN-1 domains. Like other CAMSAP proteins, PTRN-1 has three conserved regions: an N-terminal calponin homology (CH) domain, of unknown function; a central coiled-coil (CC) domain, known to interact with cytoskeleton associated proteins, and a C-terminal MT-binding CKK domain specific to CAMSAPs (*Figure 5A*). We expressed GFP-tagged fragments of PTRN-1 in the larval epidermis of *dapk-1(ju4) ptrn-1(0)* mutants as multicopy transgenes using the *dpy-7* promoter. As shown above, expression of full-length PTRN-1 rescued the *ptrn-1(0)* suppression phenotypes (*Figure 1B*). Expression of the CKK domain alone also restored the *dapk-1* epidermal morphology phenotype in *dapk-1 (ju4) ptrn-1(0)* double mutants, whereas constructs lacking the CKK domain could not rescue, suggesting the CKK domain is required for PTRN-1 function (*Figure 5B*). However, constructs containing the CKK domain and either the CC or CH domain (i.e. ΔCH or ΔCC respectively) had significantly weaker rescuing activity compared to the CKK domain alone (*Figure 5B*). These observations suggest that the CKK domain is critical for PTRN-1 function in the epidermis and is inhibited by the CH or CC domains.

Strikingly, expression of the CKK domain alone in *ptrn-1(0)* mutants caused highly penetrant Mor phenotypes, resembling those of *dapk-1(ju4)* (*Figure 5B,C*). This is in contrast to transgenic animals expressing full-length PTRN-1, which do not display such phenotypes. The CKK domain acts cell autonomously in the epidermis, as a pan-neuronal expression of the CKK domain in *ptrn-1(0)* mutants did not induce Mor phenotypes (not shown). Paralleling

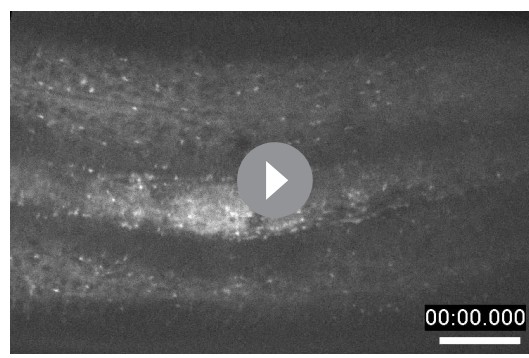

**Video 1.** EBP-GFP dynamics in adult lateral epidermis (P*col-19*).

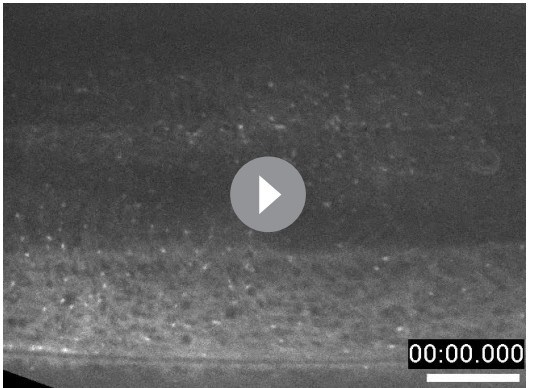

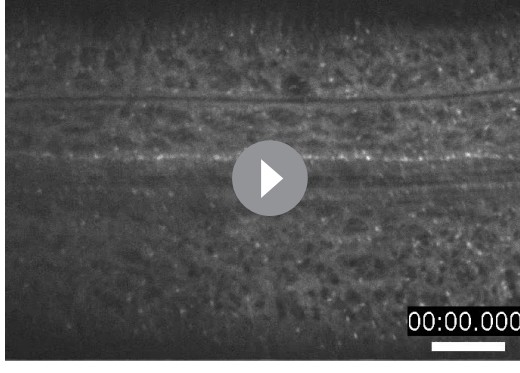

**Video 2.** EBP-GFP dynamics in adult dorsoventral epidermis.

**Video 3.** EBP-GFP dynamics in adult lateral epidermis, *dapk-1(ju4)* background.

our rescue analysis, transgenes expressing the CKK domain and the CH or CC domains did not cause epidermal defects in *ptrn-1(0)* animals. Moreover, expression of the CKK domain only caused aberrant development in a *ptrn-1(0)* background and not in a wild type background (*Figure 5B*), suggesting that CKK domain activity is inhibited by endogenous PTRN-1. In addition, expression of the CKK domain in *ptrn-1(0)* mutants induced P*nlp-29-GFP* expression, similar to that seen in *dapk-1 (ju4)* mutants (*Figure 5D*), and accelerated wound closure (*Figure 5E*). These data are consistent with DAPK-1 and PTRN-1 acting antagonistically in epidermal development, and suggest that DAPK-1 might specifically inhibit the activity of the PTRN-1 CKK domain.

## Localization of PTRN-1 along MTs, mediated by its CKK domain, correlates with defective epidermal morphogenesis

In other CAMSAP proteins the CKK domain binds MTs (*Baines et al., 2009*), whereas the CC domain confers minus-end targeting (*Goodwin and Vale, 2010*). We therefore investigated whether PTRN-1 localization correlated with its effects on epidermal morphology. Full-length GFP::PTRN-1 localized to puncta and to short filaments that were either thin (0.22 ± 0.01 μm wide) or thick (0.38 ± 0.01 μm) (*Figure 5F*); the latter co-localized with MTs (*Figure 5—figure supplement 1E*). In contrast, the CKK domain localized to longer thin filaments (*Figure 5G*; *Figure 5—figure supplement 1C*) and co-localized with MTs (*Figure 5H*). GFP::PTRN-1(ΔCKK) was almost completely punctate (*Figure 5G*). Taken together, the CKK domain is critical for localization along MTs, and MT localization is necessary but insufficient to trigger aberrant epidermal morphology (*Table 2*).

The PTRN-1 CH domain did not confer subcellular localization, whereas fragments lacking the CH domain localized to filaments and puncta, resembling full length PTRN-1 (*Figure 5G*, *Table 2*). The PTRN-1 CC domain localized primarily to puncta, whereas constructs lacking this domain (ΔCC) did not form puncta or thick filaments (*Figure 5G*). Further dissections suggest the coiled-coil subregions of the CC domain have distinct roles such that CC1 and CC2 promote localization to puncta and thick filaments (*Figure 5A*; *Figure 5—figure supplement 1A*) and inhibit the CKK domain's ability to induce epidermal morphology defects (*Figure 5—figure supplement 1B*).

We hypothesized that the expression of the CKK domain in the absence of other PTRN-1 domains causes excessive MT stabilization and aberrant epidermal development. Both full-length PTRN-1 and CKK domain transgenes restored circumferential MTs to *ptrn-1(0)* mutants (*Figure 5—figure supplement 1D*).

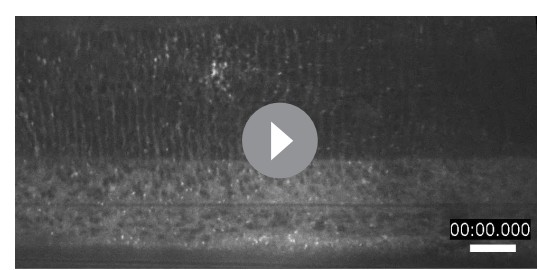

**Video 4.** EBP-GFP dynamics in adult dorsoventral epidermis, *dapk-1(ju4)* background.

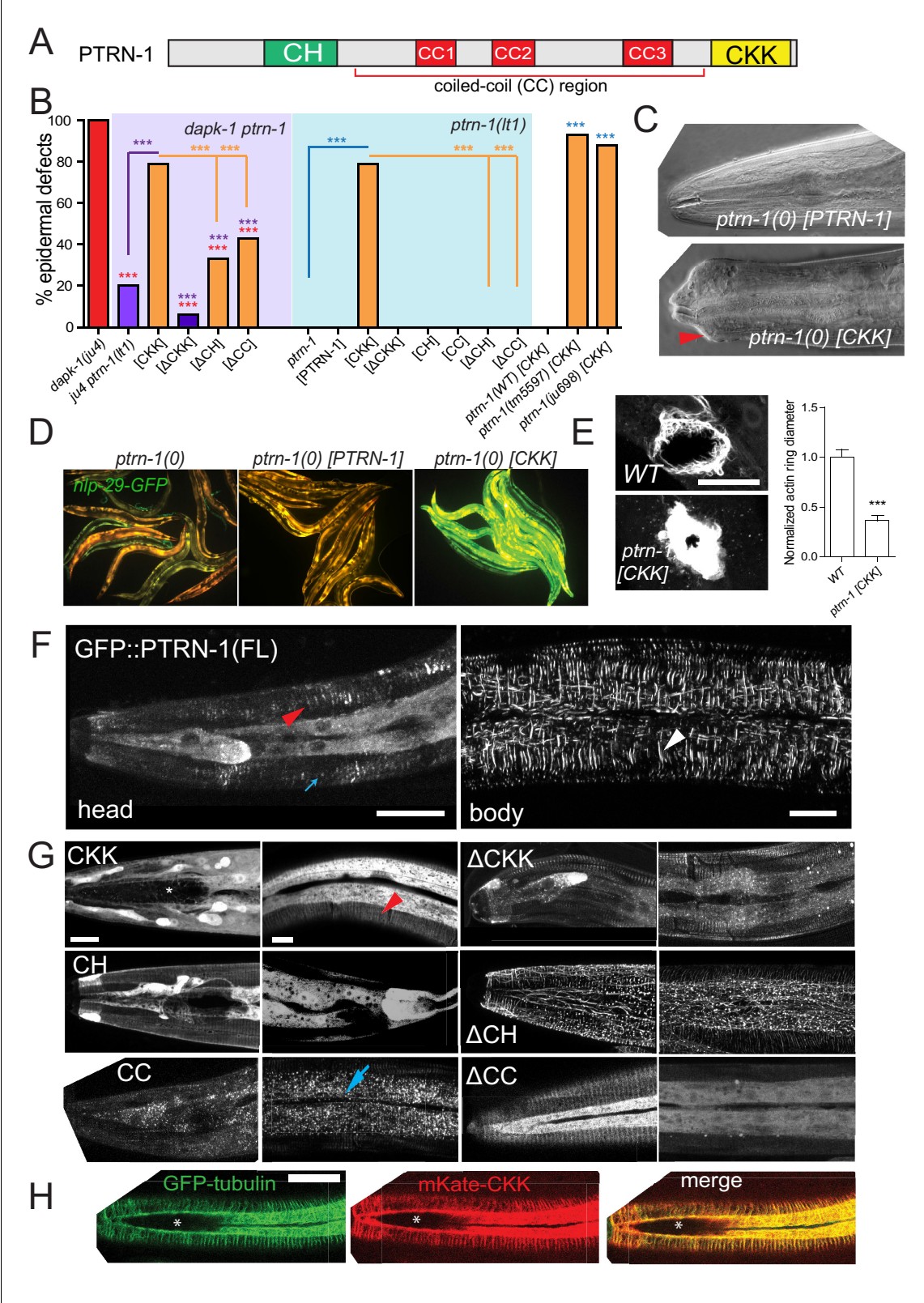

**Figure 5.** The CKK domain of PTRN-1 is required and sufficient to cause *dapk-1*-like defects in epidermal morphology. (**A**) PTRN-1 domain organization. (**B**) Quantitation of epidermal defects in animals expressing different fragments of GFP::PTRN-1. N>100. Fisher's exact test; ***p<0.001. (**C**) Representative DIC images of heads of animals expressing full length PTRN-1 or the CKK domain alone (*dpy-7* promoter, *juEx6697* and *juEx6695*), in *ptrn-1(lt1)*. (**D**) Over-expression of the CKK domain is sufficient to induce P*nlp-29*-GFP expression in *ptrn-1(0)* mutants (*frIs7; juEx7385*) and (**E**) speed up

*Figure 5 continued on next page*

*Figure 5 continued*

wound closure (*juIs352; juEx6825*). N>15, t-test; ***p<0.001. (F) Localization of full length GFP::PTRN-1 in the larval epidermis (*juEx6697*). White arrowhead points to a thick filament in the midbody lateral epidermis. PTRN-1 also localizes to puncta (blue arrow) and thin filaments (red arrowhead). All scale bars, 10 µm. (G) Localization of GFP-tagged PTRN-1 fragments in larval epidermis. (H) Colocalization of PTRN-1 CKK domain and MTs in anterior larval epidermis. Genotype: *ptrn-1(lt1); juEx6825* (P*dpy-7*-mKate2::CKK); *ltSi570* (P*dpy-7*-GFP::TBB-2). Asterisk indicates region devoid of MTs.

The following figure supplement is available for figure 5:

**Figure supplement 1.** Structure-function analyses of PTRN-1.

However, expression of the CKK domain in a *ptrn-1(0)* background caused circumferential MT bundles to be straighter than in wild type, resembling the effects of paclitaxel treatment (***Figure 4—figure supplement 1D***). *ptrn-1(0)* mutants expressing full-length PTRN-1 did not display straighter MT bundles or Mor phenotypes (***Figure 5—figure supplement 1D***). Thus the MT-stabilizing effects of the CKK domain can induce aberrant epidermal development, and are normally inhibited by other PTRN-1 domains.

## *dapk-1* mutants display increased PTRN-1 in the epidermis

The above observations suggest that in *dapk-1(ju4)* mutants, PTRN-1's MT stabilization activity, mediated by its CKK domain, might be aberrantly active. We therefore assessed whether PTRN-1 localization or expression was altered in *dapk-1(ju4)* mutants. In the wild type, PTRN-1::GFP formed thick filaments or puncta in the anterior lateral epidermis, and sparse filaments in the body epidermis (***Figure 6A***). By contrast, in *dapk-1(ju4)* adults, PTRN-1::GFP formed twice as many thick filaments in the head (***Figure 6B–E***). The increased number of PTRN-1 filaments could reflect increased PTRN-1 expression in *dapk-1(ju4)*, as multicopy GFP::PTRN-1 transgenes (***Figure 5F***) also displayed more filaments compared to those seen from a single-copy transgene. However, immunoblotting suggested overall PTRN-1 protein levels were lower in *dapk-1(ju4)* mutants compared to WT (***Figure 6—figure supplement 1A***).

Increased PTRN-1 filaments in *dapk-1* mutants might be secondary to an increase in MT number or stability. Consistent with this, paclitaxel treatment in *ptrn-1(0)* and *dapk-1(ju4) ptrn-1(0)* double mutants caused increased GFP::PTRN-1 filament formation (***Figure 6—figure supplement 1B,C***). The increased association of PTRN-1 with MTs in *dapk-1(ju4)* could reflect a positive feedback loop whereby stabilized MTs recruit more PTRN-1, resulting in further MT stabilization.

**Table 2.** PTRN-1 Structure-function analysis.

| Protein Fragment | Localization | | | Function | | |
|---|---|---|---|---|---|---|
| | Puncta | Thick filaments | Thin filaments | Restore Mor | Induce Mor | Co-loc with MTs |
| Full length | x | x | x | yes | no | yes |
| CH | - | - | - | no | no | no |
| CC | x | - | - | no | no | yes |
| CKK | - | - | x | yes | yes | yes |
| ΔCKK | x | - | - | no | no | yes |
| ΔCH | x | x | - | yes | no | ND |
| ΔCC | - | - | x | yes | no | ND |
| ΔCHCC1 | ? | x | - | ND | no | ND |
| ΔCHCC1CC2 | - | - | x | ND | slightly | ND |
| ΔCC2 | ? | x* | x | ND | no | ND |
| ΔCC3 | x | - | - | ND | no | ND |

Notes: *: Thick filaments present, but fewer compared to PTRN-1 full length or ΔCH. ?: unclear. ND: Not Determined.

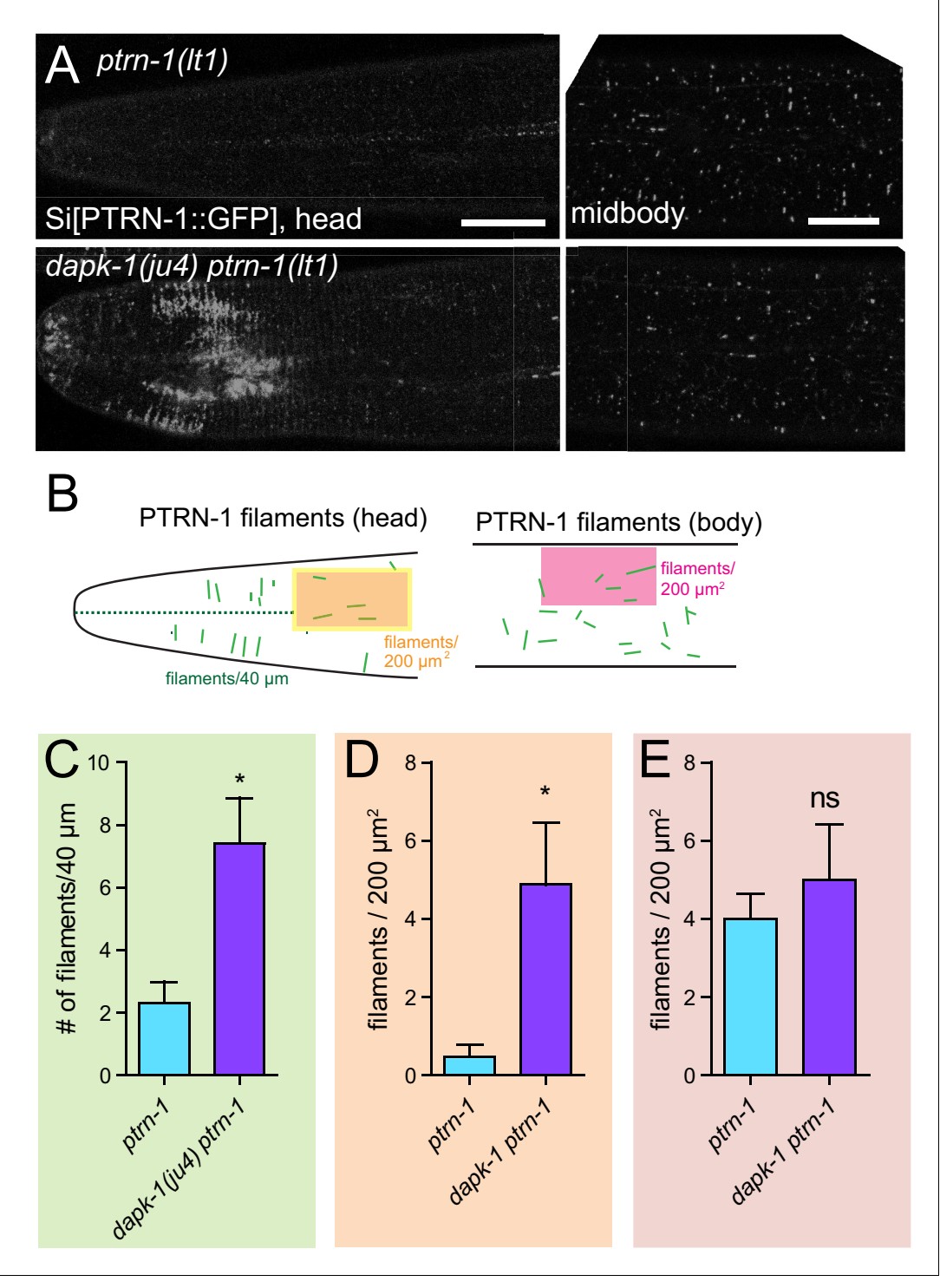

**Figure 6.** *dapk-1(ju4)* mutants display aberrant PTRN-1 localization. (**A**) Localization of PTRN-1::GFP expressed from single-copy insertion in L4 stage anterior epidermis (left) and midbody lateral epidermis (right); transgene P*dpy-7*-PTRN-1::GFP, *ltSi541*, in *ptrn-1(0)* or *dapk-1(ju4) ptrn-1(0)* background. Scale bars, 10 μm. (**B**) Schematic showing ROIs analyzed in the anterior epidermis. (**C–E**) Quantitation of PTRN-1 localization in ROIs indicated in B. Experiments performed at 25°C. N>8 animals per genotype. Bars indicate mean ± SEM. Student's t-test; *p<0.05.

The following figure supplement is available for figure 6:

**Figure supplement 1.** PTRN-1 localization is sensitive to MT polymerization and *dapk-1*.

To address how DAPK-1 might regulate PTRN-1 we examined whether DAPK-1 was capable of physical association with PTRN-1. Using co-immunoprecipitation in transfected HEK293T cells we found that PTRN-1 can be pulled down with DAPK-1 (*Figure 6—figure supplement 1D*), suggesting DAPK-1 and PTRN-1 might interact directly or as part of a complex.

## DAPK-1 undergoes directional MT-dependent transport in the epidermis

Our genetic and pharmacological analysis of the *dapk-1* mutant defect in epidermal morphology suggests DAPK-1 might inhibit MT stability, possibly via negative regulation of PTRN-1 itself. DAPK-1 might act directly or indirectly on PTRN-1. Although DAPK family members have been shown to localize to the cytoskeleton, the subcellular location of DAPK relative to MTs or PTRN-1 has not been examined. To further address whether DAPK-1 might affect PTRN-1 or MTs directly, we examined the localization of functional GFP::DAPK-1. Unexpectedly, GFP::DAPK-1 formed discrete puncta that underwent rapid, directed movement within the larval or adult epidermis (*dpy-7* promoter, *Figure 7A–D*, *Video 5*; *col-19* promoter, *Videos 6*, *7*). In the lateral epidermis GFP::DAPK-1 puncta moved along the anteroposterior body axis, occasionally reversing direction (*Figure 7C*). In contrast, puncta in the dorsoventral epidermis exhibited a bias in movement away from the lateral epidermis (*Figure 7B,D*). GFP::DAPK-1 puncta also aggregated at the edges of the epidermal ridges along the body (*Figure 7A*), as well as in the anterior epidermis, where morphological defects were prominent (*Figure 7—figure supplement 1A*).

As the pattern of GFP::DAPK-1 dynamics was reminiscent of the arrangement of the MT cytoskeleton, we assessed whether GFP::DAPK-1 dynamics were MT-dependent. In the lateral epidermis GFP::DAPK-1 puncta moved at 1.33 ± 0.028 µm/s; GFP::DAPK-1 movement in the dorsoventral epidermis was slightly slower (*Table 3*). Colchicine treatment severely reduced the number of moving GFP::DAPK-1 puncta (*Figure 7E*); movement of the remaining puncta was slightly slowed (*Figure 7—figure supplement 1B*). GFP::DAPK-1 puncta moved ~5 times faster than EBP-2::GFP comets (*Figure 7F*; *Table 3*). Moreover, in contrast to the unidirectional movement of GFP::DAPK-1 away from epidermal ridges plus-end growth occurred in both directions in the dorsoventral epidermis (*Figure 7G*). We infer that GFP::DAPK-1 dynamics reflect MT-based transport.

MT-dependent transport within the mature *C. elegans* epidermis has not been previously characterized. To understand whether other cellular components are transported within the epidermis we examined an early endosome marker, GFP::RAB-5, as endocytosis has been implicated in innate immune responses (*Dierking et al., 2011*). GFP::RAB-5 formed motile puncta that were smaller and slower-moving than those of GFP::DAPK-1, and moved in both directions in the dorsoventral epidermis (*Video 8*; *Figure 7F,G*), suggesting GFP::DAPK-1 dynamics are distinct from endosomal transport.

We asked whether DAPK-1's transport correlated with its functions in epidermal morphology. DAPK-1, like mammalian DAPK, contains an N-terminal kinase domain, a Calcium/Calmodulin binding domain, a set of ankyrin repeats, a P-loop, a cytoskeleton-interacting domain and a C-terminal death domain (*Shiloh et al., 2013*) (*Figure 1—figure supplement 1A*). Constructs lacking any one of these domains, or containing a K57A point mutation predicted to abolish kinase activity (*Deiss et al., 1995*), were unable to rescue *dapk-1(ju4)* nor did they induce *dapk-1*-like phenotypes (*Figure 7—figure supplement 1C*, *Table 4*). Uniquely, constructs lacking the kinase domain (ΔKinase) strongly enhanced the Mor phenotype of a partial loss of function allele *dapk-1(ju469)* (*Figure 7—figure supplement 1C*), and caused lethality in a *ju4* mutant background (*Table 4*). The enhancement of *dapk-1* phenotypes suggests DAPK-1(ΔKinase) inhibits DAPK-1 in a dominant-negative manner and may also inhibit parallel pathways.

We examined the localization and dynamics of these truncated proteins. Both GFP::DAPK-1 (ΔKinase) and GFP::DAPK-1(S179L) formed dynamic puncta that moved at normal or slightly slower velocities (*Table 4*; *Figure 7—figure supplement 1B*). Conversely GFP::DAPK-1 lacking either the cytoskeletal binding domain or the death domain did not form puncta (*Figure 7—figure supplement 1A*). No single domain was sufficient for the formation of motile puncta (*Table 4*). However, the kinase domain, ankyrin domain, and cytoskeleton-binding domain by themselves occasionally formed stationary puncta (*Figure 7—figure supplement 1A*). In summary, multiple non-kinase domains are required for formation of motile DAPK-1 puncta; formation of motile puncta could be necessary but not sufficient for DAPK-1 function.

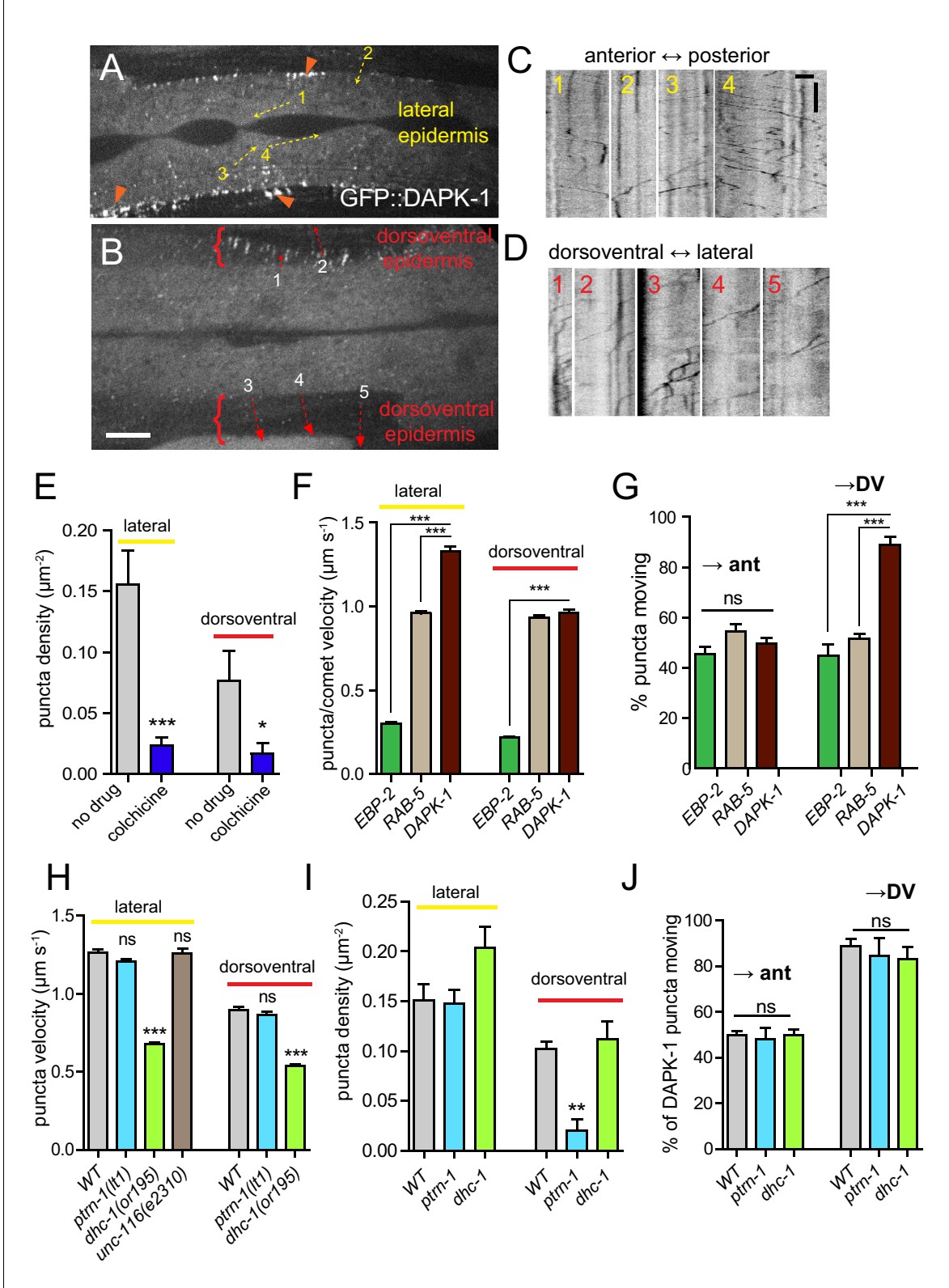

**Figure 7.** DAPK-1 undergoes MT-dependent transport in the epidermis. (**A,B**) Representative images of GFP::DAPK-1 puncta in the larval epidermis (P*dpy-7*-GFP::DAPK-1, *juEx1774*), showing lateral and dorsoventral regions, respectively. Dotted arrows indicate line scans used to generate kymographs in **C,D**. Orange arrowheads indicate clusters of GFP::DAPK-1 puncta at the boundary of the dorsoventral epidermis. Scale bars, 10 μm. (**C**) Kymographs of puncta in lateral epidermis (movie frame in A); inverted grayscale; x axis scale, 10 μm; y axis scale, 10 s. (**D**) Kymographs of puncta in

*Figure 7 continued*

dorsoventral epidermis (movie frame in B). (**E**) Density of motile GFP::DAPK-1 puncta in adult anterior lateral epidermis, with or without colchicine treatment (*cat-4*; P*col-19*-GFP::DAPK-1 *juEx4781*). N>20. (**F**) GFP::DAPK-1 puncta move faster than RAB-5 puncta or EBP-2 comets in the lateral epidermis (P*col-19*-GFP::DAPK-1, *juEx4781*). n>100. Bar charts show mean ± SEM. Statistics, Kruskal-Wallis test and Dunn's post-test; *p<0.1, ***p<0.001. (**G**) Directionality of puncta or comet motion. Approximately equal numbers of RAB-5 puncta and EBP-2 comets move in each direction; only DAPK-1 displays a significant bias in the dorsoventral epidermis. (**H**) Reduced dynein/*dhc-1* function slows GFP::DAPK-1 puncta; loss of *ptrn-1* or mutation in *unc-116*/kinesin-1 has no effect; n>100. I. *ptrn-1(0)* mutants have reduced DAPK-1 puncta in the dorsoventral epidermis. J. DAPK-1 puncta directionality is unaffected in *ptrn-1(0)* or *dhc-1* mutants.

The following figure supplement is available for figure 7:

**Figure supplement 1.** Structure function analysis of DAPK-1.

We next asked whether DAPK-1 transport required motor proteins identified as *dapk-1* suppressors. The velocity of DAPK-1 puncta did not change in *unc-116*/kinesin-1 mutants, but decreased two-fold in *dhc-1(or195)* mutants (**Figure 7H**). *ptrn-1(0)* mutants had fewer moving DAPK-1 puncta in the dorsoventral epidermis, consistent with their reduced numbers of circumferential MTs (**Figure 7I**); the velocity of the remaining GFP::DAPK-1 puncta was normal. Neither *ptrn-1(0)* nor *dhc-1* mutants affected the directional bias of GFP::DAPK-1 transport (**Figure 7J**).

## Discussion

*C. elegans* DAPK-1 acts as a negative regulator of epidermal maintenance and wound responses, through previously unknown mechanisms (*Tong et al., 2009*). Here, we identify a role for DAPK-1 in regulating the stability or the dynamics of the MT cytoskeleton. The MT minus end factor PTRN-1 is closely linked to DAPK-1 function. Moreover, DAPK-1 itself is transported along MTs, indicating a complex relationship between DAPK-1 and MT dynamics in epidermal growth and repair.

### Normal and mutant functions of DAPK-1

We screened for suppression of morphological defects of the *dapk-1(ju4)* allele, which results in a missense S179L alteration in the peptide-binding ledge of the DAPK-1 kinase domain. This allele, unlike *dapk-1(gk219)* or deletions of the *dapk-1* locus, causes fully penetrant morphological phenotypes. Our identification of stop codon mutations as intragenic revertants, as well as the finding that transgenic expression of S179L DAPK-1 induces Mor phenotypes, suggests *dapk-1(ju4)* is a gain of function mutation.

How the S179L change causes a gain of function remains to be determined. One hypothesis for why *dapk-1(ju4)* has a more penetrant effect on epidermal morphology than the complete loss of *dapk-1* function is that the S179L change alters the phosphorylation specificity of DAPK-1. Like mammalian DAPK family members, DAPK-1 likely has many phosphorylation targets, some of which may affect epidermal morphology. One clue to the effect of S179L on protein function comes from our structure function study. Truncated versions of mammalian DAPK1 can have dominant-negative behavior (*Cohen et al., 1997*, *1999*; *Lin et al., 2007*). In our assays, DAPK-1 (Δkinase) strongly enhanced, but did not induce, *dapk-1* phenotypes, while DAPK-1 (kinase dead) did neither. Thus, DAPK-1(S179L) is unlikely to be kinase-dead, but may have an aberrantly regulated kinase activity. Inappropriate activity of DAPK-1 targets in *dapk-1(ju4) mutants* may account for the stronger morphological defects in this genetic background.

Our pharmacological data suggest DAPK-1's normal function is to restrain epidermal MT

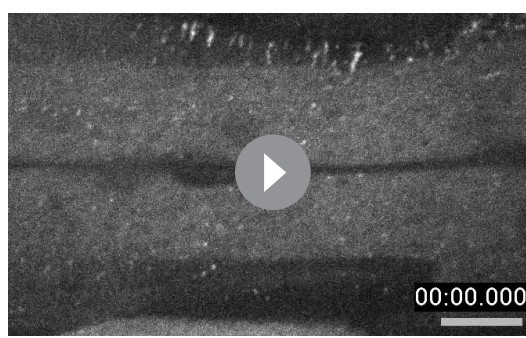

**Video 5.** GFP::DAPK-1 dynamics in larval epidermis. Lateral seam cells are in the center of the movie.

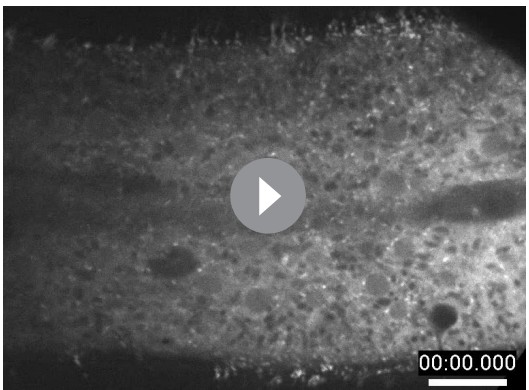

**Video 6.** GFP::DAPK-1 dynamics in adult lateral epidermis. Lateral seam is in center.

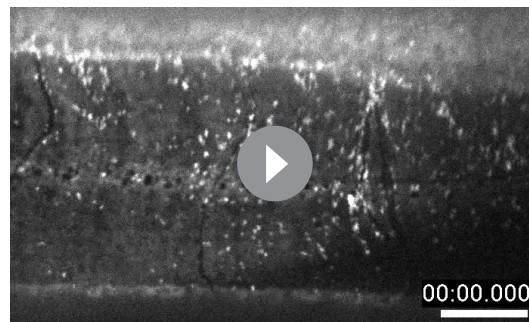

**Video 7.** GFP::DAPK-1 dynamics in adult dorsoventral epidermis. Dorsal midline is at bottom of movie.

stabilization. Our suppressor screen and tests of candidate genes suggests a specific interaction between *ptrn-1* and *dapk-1*; thus, DAPK-1 may be regulating epidermal MT stability at least partially through PTRN-1. In addition, we identified *dhc-1* and other cytoskeletal associated proteins that specifically modify *dapk-1* morphological phenotypes; such genes may act downstream of DAPK-1.

Upregulation of the epidermal innate immune response observed in *dapk-1(ju4)* is genetically separable from epidermal morphology defects (*Tong et al., 2009*) (*Figure 2A,B*). Our analysis using MT drugs shows that the concentrations of drugs sufficient to suppress or induce morphological defects are insufficient to suppress or induce innate immune responses, consistent with previous observations (*Zhang et al., 2015*). Thus, epidermal morphology is more sensitive to MT disruption than is the innate immune response, and MT stabilization may affect innate immune responses indirectly via effects on epidermal integrity.

## PTRN-1's function in epidermal development and the MT cytoskeleton

The Patronin/CAMSAP family of MT minus end-binding proteins has become the focus of intense investigation in several organisms (*Akhmanova and Hoogenraad, 2015*). The sole *C. elegans* member of this family, PTRN-1, is dispensable for epidermal development in part due to a parallel pathway involving γ-tubulin and the Ninein-related protein NOCA-1 (*Wang et al., 2015*). *ptrn-1(0)* single mutant larvae show grossly wild-type epidermal MT plus-end dynamics and have slightly fewer circumferential MT bundles in the epidermis (*Wang et al., 2015*), a phenotype that worsens in adults (this work). Thus during postembryonic development the epidermis becomes more dependent on PTRN-1 and on noncentrosomal MT arrays. Although *ptrn-1* is largely dispensable for epidermal morphology, its requirement becomes evident in wound repair in adults, paralleling the role of

**Table 3.** Dynamics parameters in the epidermis.

| Type of Dynamics | Transgene | Velocity (mean ± SEM) μm s$^{-1}$ | | Directionality | |
|---|---|---|---|---|---|
| | | Lateral | Dorso-ventral | Lateral | Dorso-ventral |
| MT plus-end growth | P*col-19*-EBP-2::GFP | 0.30 ± 0.00 | 0.22 ± 0.007 | A→P P→A | Lat→DV DV→Lat |
| Early endosome transport | P*dpy-7*-GFP::RAB-5 | 0.96 ± 0.014 | 0.93 ± 0.014 | A→P P→A | Lat→DV DV→Lat |
| DAPK-1 | P*dpy-7*-GFP::DAPK-1 | 1.33 ± 0.028 | 0.96 ± 0.021 | A→P P→A | Lat→DV |
| DAPK-1 | P*col-19*-GFP::DAPK-1 | 1.26 ± 0.02 | 0.90 ± 0.017 | A→P P→A | Lat→DV |

Notes: A→P: Anterior to Posterior. P→A: *vice versa*. Lat→DV: Lateral to dorsoventral epidermis. DV→Lat: *vice versa*.

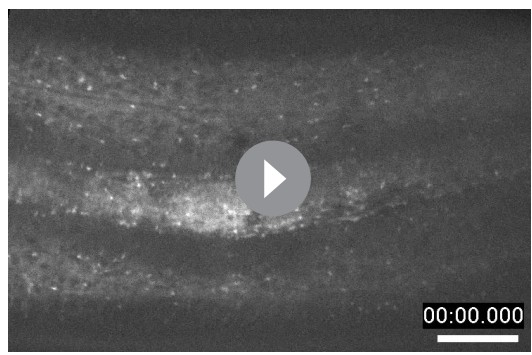

**Video 8.** P*dpy-7*-GFP::RAB-5 dynamics.

PTRN-1 in adult axon regeneration in *C. elegans* (*Chuang et al., 2014*). The requirement for PTRN-1 in repair but not in development may reflect the prominent role of noncentrosomal MT arrays in mature neurons and epithelia.

The MT-binding PTRN-1 CKK domain is not only sufficient to rescue *ptrn-1(0)* suppression phenotypes, but can induce epidermal defects when overexpressed. Depending on the particular CAMSAP family member, the CKK domain can target MT minus ends, or can bind along the lattice (*Goodwin and Vale, 2010*; *Jiang et al., 2014*). In *C. elegans* PTRN-1 requires the CKK domain to localize to MTs in neurons and muscles (*Richardson et al., 2014*), and the CKK domain can stabilize neuronal MTs (*Chuang et al., 2014*), consistent with MT stability being a determinant of epidermal morphology.

Little is known about how CKK domain activity is regulated, but our data suggest DAPK-1 might directly or indirectly inhibit CKK activity; moreover, the CH or CC domains of PTRN-1 appear to inhibit CKK activity. As the CC domain of PTRN-1 appears to target to minus ends, a possible mechanism is that in the absence of the CC domain, ectopic association of the CKK domain along the MT lattice (long thin filaments) results in aberrant MT stabilization. DAPK-1 might inhibit the activity of the CKK domain indirectly via the CC or CH domains (*Figure 8*).

## DAPK-1 and MT-dependent transport in the epidermis

To our knowledge, DAPK family members have not been previously shown to undergo MT-dependent transport. Many questions remain concerning the mechanism of DAPK-1 transport; here we speculate on the possible function of DAPK-1 motility in the *C. elegans* epidermis, and why aberrant DAPK-1 function should trigger altered epidermal morphology.

In the *C. elegans* epidermis the MT cytoskeleton is required for cell shape changes (*Priess and Hirsh, 1986*; *Quintin et al., 2016*) and for nuclear migrations (*Starr, 2011*). Epidermal MTs are critical for larval development: noncentrosomal arrays of the larval epidermis require PTRN-1 and the NOCA-1/γ-tubulin pathway (*Wang et al., 2015*). In contrast, PTRN-1 is required non-redundantly for *dapk-1* epidermal defects, suggesting that in this context PTRN-1 and NOCA-1 pathways are not equivalent.

**Table 4.** DAPK-1 structure-function analysis.

| Protein Fragment | Puncta? | Motile? | Rescues Mor* | Induces Mor[†] | Enhances Mor[‡] | Lethal in *dapk-1(ju4)* |
|---|---|---|---|---|---|---|
| Full Length | yes | yes | yes | no | no | no |
| Δ Kinase | yes | yes | no | no | yes | yes |
| Δ Kinase + CaM Bind Domain (BD) | yes | yes | no | no | yes | yes |
| Δ Cytoskeletal BD | no | no | no | no | no | no |
| Δ Death Domain + C terminus | no | no | no | no | no | no |
| Kinase only | some | no | no | no | ND | ND |
| CaM BD only | no | no | no | no | ND | ND |
| Ankyrin Domain only | some | no | no | no | no | no |
| Cytoskeletal BD only | some | no | no | no | no | no |
| Mid + DD + Ct | no | no | no | no | no | no |
| S179L *dapk-1(ju4)* | yes | yes | no | yes | yes | yes |
| K57A (kinase dead) | yes | yes | no | no | no | no |

Notes: *: In *dapk-1(ju4)*. [†]: In WT background. [‡]: In *dapk-1(ju469)*.

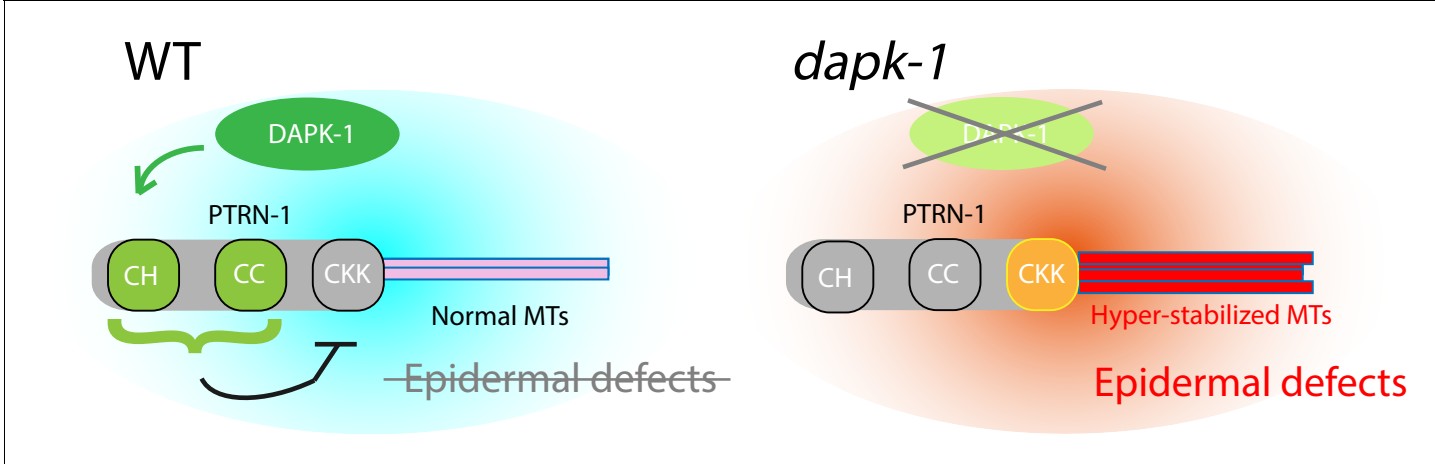

**Figure 8.** Model for DAPK-1-mediated regulation of epidermal MTs via PTRN-1.

A key question is why MT stabilization should cause progressive cuticle hypertrophy in specific regions of the epidermis. We speculate that the architecture of the epidermis may make it increasingly dependent on intracellular transport as it grows during larval development and adult life. Epidermal nuclei are confined to lateral and ventral ridges. The squamified dorsoventral epidermis is a barrier to diffusion. Thus, as in neuronal axons, extensive intracellular transport may be required to supply distant regions of the cell with cellular constituents. DAPK-1 may maintain MT dynamics required for active transport of materials to the remoter areas of the epidermis. Furthermore, our data show that the loss of *dapk-1* function has a stronger effect on MT architecture in the lateral epidermis compared to the dorsoventral epidermis, perhaps due to differences in the local MTOCs or PTRN-1 in these different compartments. Cell type-specific differences in CAMSAP function have been recently been reported in Drosophila. The spectraplakin Shot and Drosophila CAMSAP Patronin co-localize, maintain MT arrays, and act to generate cell polarity; however, in the oocyte, Shot and PTRN-1 appear to act in the same pathway (*Nashchekin et al., 2016*), whereas in follicular epithelial cells they appear to act in parallel (*Khanal et al., 2016*). Possibly, PTRN-1 or non-centrosomal MTOCs in the lateral epidermis are more dependent on DAPK-1 function than in the dorsoventral epidermis.

In conclusion, we have revealed an unexpected regulatory interaction between DAPK-1 kinase and the MT cytoskeleton in epidermal development and wound responses. Many questions remain to be explored, especially whether PTRN-1 or other MT-associated proteins are direct substrates of DAPK-1. More broadly, the mechanisms and roles of intracellular transport within the epidermis could be a model for intracellular transport in other syncytial tissues.

## Materials and methods

### *C. elegans* genetics

Strains were maintained at 20–22.5°C on NGM agar plates with *E. coli* OP50, unless otherwise indicated. Epidermal morphology defects (Mor phenotypes) were scored in animals grown at 20°C unless stated otherwise. Strains were constructed using standard procedures, and were genotyped by PCR or sequencing. Previously described mutations include: *dapk-1(ju4, gk219, ju469)*, *ptrn-1 (tm5597, lt1)*. New *dapk-1* suppressor mutations and additional candidate genes tested are listed in *Table 1*. New transgenic strains are listed in *Supplemental file 1A*. Representative extrachromosomal arrays are listed. *dapk-1* CRISPR deletions were generated by injecting 27 μM circular RNAs (crRNA) CACCATTAACATTCTCTCTC and ATACGAACGCATAATAA, *dpy-10* crRNA as a co-injection marker, tracrRNA, and purified Cas9 protein (QB3 MacroLab, UC Berkeley), following published methods (*Paix et al., 2015*).

## Molecular biology

New plasmids were made by Gateway cloning or by Gibson isothermal assembly. Site directed mutagenesis of the *dapk-1* cDNA used Gibson assembly. Mutant versions of *ptrn-1* cDNAs have been described (*Chuang et al., 2014*). New plasmids are listed in *Supplemental file 1B*.

## *dapk-1* suppressor screens, mapping, and whole genome sequencing

We mutagenized *dapk-1(ju4)* animals with 47 mM EMS using standard protocols and screened the $F_2$ progeny (~5000 haploid genomes) for suppression of Mor phenotypes. *dapk-1(ju4)* suppressors were outcrossed once prior to extraction of genomic DNA and whole genome sequencing (BGI Americas). Sequence data were processed and analyzed using MAQgene (*ju697, ju698*) or a Galaxy workflow, comparing datasets to each other and to the wild type reference sequence in WS220. We used SNP mapping to locate suppressor *ju698* to a 2.2 Mb interval within which we identified a stop codon mutation in *ptrn-1*. We mapped *ju697* to chromosome I, where we identified a missense alteration in *dhc-1*.

## Microtubule drug treatment

Solutions of drugs in M9 were pipetted onto small NGM agar plates (final concentrations 1–5 mM colchicine, 5–15 µM paclitaxel, all from Sigma-Aldrich, St Louis, MO). Plates were dried for 24 hr, then seeded with *E. coli* OP50. Two days later, the embryos were plated and allowed to develop at 20°C. Three days later, young adult animals were scored for epidermal defects or assayed for antimicrobial reporter gene expression. To visualize MTs after drug treatment, L4 animals were anesthetized in 0.5 mM levamisole and 2 µM paclitaxel or 2 mM colchicine solution for 2 hr before imaging.

## Wounding assay

Needle wounding and actin ring quantitation were performed as described (*Xu and Chisholm, 2011*).

## Imaging and analysis of epidermal protein dynamics

We found that GFP::TBB-2, EBP-2::GFP, and GFP::DAPK-1 dynamics were inhibited by several anesthetics used to immobilize *C. elegans* for imaging, including muscimol, 1-phenoxy-2-propanol, and levamisole. GFP::DAPK-1 dynamics were also highly sensitive to physical stress; for example, immobilization in 10% agarose and 0.05 µm beads (*Kim et al., 2013*) immediately blocked dynamics. For the analyses reported here we immobilized animals in 0.5 mM levamisole solution for 50 min, then imaged dynamics in the next 30 min.

We imaged subcellular dynamics using a custom-built microscope (Solamere Technologies) with a Yokogawa CSU-X spinning disk confocal head, a Photometrics Cascade II EM-CCD camera, and a Zeiss NA 1.46 objective. Images were acquired at 4.2 Hz, with exposure time 114 ms at 1000 x magnification, for 100 or 200 frames. We generated kymographs using Metamorph software and determined particle velocity by manual measurement of track angles. We manually counted moving puncta or comets in a 10 frame window, using a 200 µm$^2$ region of interest (ROI) in the lateral epidermis or a 300 µm$^2$ ROI in the dorsoventral epidermis.

## Western blotting and co-immunoprecipitation

Worm protein samples were prepared by boiling 10 µl pellets of mixed-stage worms in SDS-mercaptoethanol solution. We detected PTRN-1 using antibody OD208A (*Wang et al., 2015*); we used the anti-actin antibody ACTN05 (Abcam, Cambridge, MA, C4) as a loading control. We used HRP-linked anti-rabbit IgG NA934V and anti-mouse NXA931 as secondary antibodies (GE Healthcare Lifesciences) at 1:1000 dilution in TBS, and added SuperSignal West Pico Chemiluminescent substrate (Thermo Fisher Scientific, Waltham, MA). The blot result was replicated once.

To test co-immunoprecipitation, a 1:1 ratio of tagged DAPK-1 and PTRN-1 were co-transfected using Lipofectamine 2000 (Invitrogen) into HEK293 cells growing on poly-D-Lysine-coated plates (Sigma-Aldrich) in Opti-MEM (Thermo Fisher Scientific). After two days, cells were collected in cold PBS and lysed in lysis buffer (50 mM Tris-Cl pH 7.4, 150 mM NaCl, 1% NP40, 0.5 mM EDTA, 3 mM MgCl$_2$) for 30 min at 4°C. M2-FLAG conjugated magnetic beads (Sigma) were used for IP, and

mouse anti FLAG (F1807, Sigma, 1:1000) and anti HA (HA-7, Sigma, 1:5000) antibodies were used for western blotting. The co-IP was repeated twice.

## Analysis of antimicrobial peptide reporter expression

For visual comparisons of P*nlp-29*-GFP/P*col-12*-dsRed (*frIs7*) expression we took images on a Zeiss compound microscope using a GFP long-pass filter set. To quantify P*nlp-29*-GFP (*frIs7*) expression levels we used the COPAS *C. elegans* BIOSORT (Union Biometrica, Holliston MA) at the Troemel lab (UCSD). We calculated the ratio of green fluorescence to the internal control P*col-12*-dsRed.

## Imaging and quantitation of epidermal MT and PTRN-1 distribution

Quantitation of MT bundles (P*col-19*-GFP::TBB-2, *juSi239*): (1) We calculated the anisotropy index using the ImageJ plug-in FibrilTool (*Boudaoud et al., 2014*) analyzing a rectangular ROI of 100 µm$^2$ in the anterior lateral epidermis. (2) We measured MT bundle length by drawing a line scan along a circumferential MT, from the edge of the lateral epidermis or the dorsal or ventral midline, to the end of the bundle. MT bundles detached from the epidermal ridges were not measured. (3) We counted MT bundles in a 40 µm ROI along the lateral epidermis along the anteroposterior axis. (4) We counted MT bundles crossing a 40 µm line in the dorsoventral region, including bundles not attached to the lateral or dorsoventral networks. For parameters 2–4 the ROI begins 40 µm behind the nose tip and extends 40 µm posteriorly. Schematics of ROIs are in *Figure 4B*.

Quantitation of GFP::PTRN-1 filaments (P*dpy-7*-GFP::PTRN-1, *ltSi541*): (1) Number of filaments in the dorsoventral regions of the head were counted in an ROI extending 40 µm from the nose tip (*Figure 6B*). (2) Filaments in the lateral head epidermis were counted in a 200 µm$^2$ ROI, beginning 40 µm behind the nose. (3) Filaments were counted in the anterior lateral epidermis in a 200 µm$^2$ ROI. A line of GFP > 1 µm was considered a filament. We did not examine the defective region to avoid quantifying changes in localization secondary to the aberrant epidermal morphology.

## Statistics

All statistical analyses were performed with GraphPad Prism.

# Acknowledgements

We thank our lab members for advice and discussion, Yishi Jin and Nathalie Pujol for comments on the manuscript, and the *Caenorhabditi*s Genetics Center (CGC) and the Mitani lab for strains. We thank Shaohe Wang and Karen Oegema for discussions and reagents, Kirthi Reddy and Emily Troemel for use of their COPAS worm sorter, and Abby Dernburg for advice on CRISPR/Cas9 methods. We thank Salvatore Cherra for developing the Galaxy workflow, and Laura Toy for generating *dapk-1* CRISPR alleles. MC was supported by the UCSD Cellular and Molecular Genetics Training Grant (NIH T32 GM007240) and by the UCSD Frontiers of Innovation Scholars Program. This work was supported by NIH R01 GM054657 to ADC.

# Additional information

## Funding

| Funder | Grant reference number | Author |
|---|---|---|
| National Institute of General Medical Sciences | T32 GM007240 | Marian Chuang |
| National Institute of General Medical Sciences | R01 GM054657 | Amy Tong Suhong Xu Andrew D Chisholm |
| Junior Thousand Talents Program of China | | Suhong Xu |

The funders had no role in study design, data collection and interpretation, or the decision to submit the work for publication.

## Author contributions
MC, TIH, AT, SX, Conception and design, Acquisition of data, Analysis and interpretation of data; ADC, Conception and design, Analysis and interpretation of data, Drafting or revising the article

## Author ORCIDs
Andrew D Chisholm, http://orcid.org/0000-0001-5091-0537

## Additional files

### Supplementary files
• Supplementary file 1. Newly generatedstrains and plasmids. (A) List of new strains and genotypes. (B) List of new plasmids.

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
