## [Decision Letter]

Thank you for submitting your article "DAPK interacts with Patronin and the microtubule cytoskeleton in epidermal development and wound repair" for consideration by *eLife*. Your article has been favorably evaluated by Fiona Watt as the Senior Editor and four reviewers, one of whom is a member of our Board of Reviewing Editors.

The reviewers have discussed the reviews with one another and the Reviewing Editor has drafted this decision to help you prepare a revised submission. We hope you will be able to submit the revised version within two months, so please let us know if you have any questions first.

Summary:

In this report, Chuang et al. explore the role of the death associated protein DAPK-1, a structurally unique 160-kD calmodulin dependent serine-threonine kinase that has been implicated in a range of processes in mammals, using *C. elegans* epidermal cells as a model. Through a suppressor screen, they uncover that DAPK-1 functions through the microtubule cytoskeleton and more specifically the minus end binding protein patronin. The authors present rigorous genetic and pharmacological evidence suggesting that the epidermal defects associated with aberrant DAPK-1 function are due to hyperstabilization of the microtubule cytoskeleton caused by misregulation of patronin.

Essential revisions:

Both the *ju4* (gain of function) and *gk219* (loss of function) alleles are rescued by mutations in patronin and *dhc*. The interchangeable use of these alleles (and their ability to act similarly) is not well explained. Does the deletion mutant have the same microtubule morphology defects as the gof? It is necessary to fill in the gaps in experiments using *gk219* for all experiments, as the nature of this allele is understood.

Of course, if you have data on physical interaction between DAPK and Patronin, these would greatly help the paper.

---

## [Author Response]

*Both the ju4 (gain of function) and gk219 (loss of function) alleles are rescued by mutations in patronin and dhc. The interchangeable use of these alleles (and their ability to act similarly) is not well explained. Does the deletion mutant have the same microtubule morphology defects as the gof? It is necessary to fill in the gaps in experiments using gk219 for all experiments, as the nature of this allele is understood.*

We use two different alleles, *ju4*, which displays 100% penetrance of the Mor phenotype and is easiest to score, and *gk219*, which has partial penetrance, allowing us to survey for enhancement as well as suppression of the phenotype. We believe *gk219* represents the null phenotype of *dapk-1*, as explained below. The *ju4* missense alteration is a gain of function that appears to have dominant-negative activity, as the phenotypes overall resembles a more penetrant version of the null phenotype.

As requested, we have repeated some key experiments with the *gk219* allele. We examined MT arrays in *gk219* and in *gk219 ptrn-1* mutants, and show the data in Figure 4—figure supplement 1. In summary, changes in MT arrays in these mutants are not significantly different from *ju4* and *ju4 ptrn-1* mutants. We had previously performed pharmacological tests on *gk219*, as shown in Figure 3 and Figure 3—figure supplement 1, and found colchicine can suppress the Mor phenotype in both *dapk-1* alleles. We now have also added the wound closure data on *gk219* and *gk219 ptrn-1* mutants. Overall, the morphological, cytoskeletal and wound healing defects of *gk219* mutants are either similar to or slightly weaker than those of *dapk-1(ju4)*.

In addition, to address the nature of the *dapk-1(gk219)* deletion allele, we used CRISPR/Cas9 to generate deletions of the entire *dapk-1* transcription unit (23 kb) (Figure 1—figure supplement 1). Three such deletions tested resemble *gk219* in penetrance of morphological defects, consistent with *gk219* causing complete loss of function.

We explored the nature of the *dapk-1(ju4)* mutation by structure-function analysis, as shown in Figure 7—figure supplement 1 and in Table 3. None of the truncations tested, nor kinase dead DAPK-1, mimics the effects of *dapk-1(ju4)/*DAPK-1(S179L). We speculate that the S179L alters how the DAPK-1 kinase interacts with substrates and have expanded our discussion of this point (subsection “Normal and mutant functions of DAPK-1”).

*Of course, if you have data on physical interaction between DAPK and Patronin, these would greatly help the paper.*

To address this point we have now performed co-immunoprecipitation experiments in HEK293T cells. As shown in Figure 6—figure supplement 1, after immunoprecipitation of Flag-DAPK-1 we were able to detect HA-PTRN-1 (2 replicates), indicating these proteins can associate with each other.